# Parental satisfaction towards care given at neonatal intensive care unit in Ethiopia: A systematic review and meta-analysis

Teklehaimanot Gereziher Haile[1]*, Gebreamlak Gebremedhn Gebremeskel[2], Teklewoini Mariye[2], Guesh Mebrahtom[2], Woldu Aberhe[2], Abrha Hailay[2], Teklehaymanot Huluf Abraha[3], Negasi Asres[4], Teklay Guesh[4], Girmay Teklay[5]

1 Department of Maternity and Neonatal Nursing, School of Nursing, College of Health Sciences and Comprehensive Specialized Hospital, Aksum University, Aksum, Tigray, Ethiopia, 2 Department of Adult Health Nursing, School of Nursing, College of Health Sciences and Comprehensive Specialized Hospital, Aksum University, Aksum, Tigray, Ethiopia, 3 Department of Reproductive and Family Health, School of Public Health, College of Health Sciences and Comprehensive Specialized Hospital, Aksum University, Aksum, Tigray, Ethiopia, 4 Department of Epidemiology and Biostatistics, School of Public Health, College of Health Sciences and Comprehensive Specialized Hospital, Aksum University, Aksum, Tigray, Ethiopia, 5 Department of Pediatric and Child Health Nursing, School of Nursing, College of Health Sciences and Comprehensive Specialized Hospital, Aksum University, Aksum, Tigray, Ethiopia

* teklehaimg@gmail.com

**Data Availability Statement:** All relevant data are within the manuscript and its Supporting information files.

## Abstract

### Background

Satisfaction reflects a consumer's judgment regarding the quality of care received. Satisfaction of parents with healthcare services can serve as a reliable indicator for evaluating the quality of care given to their newborns. The aim of this study is to determine the pooled prevalence of parental satisfaction with neonatal intensive care unit services and its associated factors in Ethiopia.

### Methods

International databases such as PubMed/Medline, African Journals Online, SCOPUS, Web of Science, and Google Scholar were used to identify relevant published articles. Risk of bias was assessed independently by two authors. Funnel plot and Egger regression test were also employed to assess publication bias. A significant p-value of 0.05 or less suggests the presence of publication bias. PROSPERO registration number is CRD42024570971.

### Results

The researchers initially identified a total of 1,264 articles through their search strategy from all databases. Finally, eleven studies met the inclusion criteria and were considered suitable for the final meta-analysis. The pooled prevalence of parental satisfaction with neonatal intensive care unit services in Ethiopia was 55% (95% CI: 49%–60%). There was neither publication bias nor small study impact among the included studies. Based on the subgroup analysis by region, the pooled prevalence in Oromia were slightly higher, 63% (95% CI: 57%–69%). Birth weight, length of hospital stays, place of residence, and availability of

**Funding:** The author(s) received no specific funding for this work.

**Competing interests:** The authors have declared that no competing interests exist.

**Abbreviations:** AOR, Adjusted Odds Ratio; CI, Confidence Interva; NCU, Neonatal Care Unit; NICU, Neonatal Intensive Care Unit; PRISMA, Preferred Reporting Items for Systematic reviews and Meta-Analysis; WHO, World Health Organization.

necessary information are factors identified as significant predictors for parental satisfaction with neonatal intensive care unit service.

## Conclusion

According to the study's findings, there is a lower pooled prevalence of parental satisfaction than in previous research that was published in some countries. By working to raise the satisfaction levels of parents in neonatal intensive care units, healthcare facilities can enhance the overall experience for both parents and infants, leading to improved outcomes and a more positive care environment.

## Introduction

Consumer satisfaction is a measure of how well care is perceived. In today's consumer-oriented society, measuring satisfaction with care has become an essential part of evaluating health care services. The neonatal period, which spans from birth to 28 days of age, is the most vulnerable to changes in the infant's health and survival. In Neonatal Care Unit (NCU) situations, infants are unable to express their health needs and satisfaction. Meanwhile, World Health Organization (WHO) advocated respectful care for both infants and families in NCUs and emphasized supporting parents' participation in infant care [1–4]. Patient satisfaction is regarded as an essential element in assessing healthcare quality. Consequently, the satisfaction of parents with healthcare services can serve as a reliable indicator for evaluating the quality of care given to their newborns [5].

The World Health Organization classifies a preterm newborn as one who is born before the full 37 weeks of gestation. The annual number of preterm neonates born is estimated to be 15 million, and this figure is rising. Preterm newborns may have high levels of organic immaturity due to disruptions in the gestational cycle; this condition may sometimes require a baby to be admitted to a neonatal intensive care unit. Preterm delivery is the leading cause of neonatal mortality in the first four weeks of life related complications that claim the lives of over a million babies annually [6–9]. Current developments in the medical care of sick and premature infants are reflected in the Neonatal Intensive Care Unit (NICU). Approximately 7 newborns are admitted to a NICU for every 100 live births [10].

In Neonatal Intensive Care Units (NICUs), evaluating parent satisfaction and their experiences becomes crucial for assessing clinical practices and enhancing the care provided to infants and parents [11–13]. The emotional and stress levels are particularly heightened when a baby is born prematurely or with health issues and requires admission to a NICU. Parental stress is often linked to concerns about the baby's health, their overall well-being, and the adjustments required in their parenting role. Feelings of grief due to the loss of an expected healthy child are also common [14–17]. Additionally, it is essential for healthcare staff to deliver care that acknowledges the needs and experiences of the entire family [18, 19].

According to the WHO fact sheets from 2021, the neonatal period accounts for 47% of all deaths among children under the age of five, an increase compared to the percentage in 1990 (40%). Annually, approximately one million infants pass away on the day they are born, and an additional two million die within the first week of life. Sub-Saharan Africa has the highest neonatal death rate in the world, with 27 deaths per 1000 live births, followed by southern Asia with 23 deaths per 1000 live births [20, 21]. Complications and morbidity in neonates are often attributed to a lack of quality care, specifically parental dissatisfaction with the care

provided in NICUs, as well as insufficient professional care and treatment [22]. This leads to inefficiencies in infant care due to a shortage of adequate nursing, medical attention, and family-centered care [23].

In Ethiopia, the mortality rate stands at 29 deaths per 1000 live births [24], which is one of the highest rates in Africa. To address this issue and work towards achieving the Sustainable Development Goals (SDGs), the Federal Ministry of Health (FMOH) in Ethiopia has placed significant emphasis on expanding high-quality and impactful interventions for newborns in health centers and hospitals. This includes the establishment of basic newborn care units, known as "newborn corners," in health centers and NICUs in hospitals. As an initial step, the FMOH, in collaboration with its partners, has begun the process of enhancing and/or establishing NICUs in selected federal and university teaching hospitals [25].

Identifying and prioritizing areas of care can be facilitated by understanding the aspects that contribute to satisfaction. When parents are satisfied with the care they receive their babies, promotes acceptance of the care, fosters a trusting relationship with healthcare providers, encourages active involvement in infant care, and ultimately leads to improved health outcomes [1, 26]. Furthermore, adopting a family-centered care approach has several benefits, including reducing the length of hospital stays, alleviating parental stress and anxiety upon discharge, and reducing the rate of readmissions to the NICU. Consequently, this approach enhances parental satisfaction with the services provided [27].

Reducing newborn mortality is a key objective for the WHO due to its significance as a public health concern. A crucial initial step in achieving this goal is to assess the satisfaction of parents with neonatal care. In Ethiopia, efforts have been made to enhance the quality of the healthcare system through various programs, training healthcare staff, and educating the general public about healthcare-seeking behaviors. This study, conducted on parental satisfaction with care given at the NICU and its associated factors in Ethiopia, and valuable to the extension of the available evidence. Therefore, data from systematic reviews and meta-analyses on parental satisfaction can be valuable in improving care and parental support in NICUs, informing policymakers, administrators, and providing recommendations for future clinical practice and guidelines to enhance the overall quality of care provided at the NICU.

## Methods and materials

### Study design, method of search and data sources

A systematic review and meta-analysis were conducted to examine the pooled prevalence of parental satisfaction towards care given at NICU and its associated factors in Ethiopia. The study included both published and unpublished articles, ensuring a comprehensive review of the available literature on this topic. This systematic review and meta-analysis were conducted in accordance with the 2020 Preferred Reporting Items for Systematic Reviews and Meta-Analyses (PRISMA) [28] (S1 Table). This systematic review and meta-analysis is registered in PROSPERO with a registration number of CRD42024570971. A comprehensive search was conducted in international databases such as PubMed/Medline, African Journals Online, SCOPUS, Web of Science, and Google Scholar to identify relevant published articles. Additionally, grey literature was used to include unpublished articles.

The following search terms were used with the Boolean operators AND, and OR to retrieve articles: (woman OR mother OR father OR parent) AND satisfaction AND NICU AND NICU services AND ("mother's satisfaction" OR "maternal satisfaction" "father's satisfaction" OR "parental satisfaction") AND ("NICU services" OR "neonatal care") AND ("associated factors" or "magnitude" or "predictors" or "prevalence" or "incidence" or "risk factors" or "determinants") AND using each name of Ethiopia regions.

## Study selection

For the meta-analysis, studies conducted in Ethiopia that reported on the prevalence of parental satisfaction and its associated factors were selected. Endnote X8 software was used to screen articles in the initial phase. Duplicate files were eliminated after exporting all articles from various databases to Endnote X8. Three researchers (GGG, TM, and GM) reviewed the remaining articles and abstracts to determine their eligibility for full-text appraisal. In case of any discrepancies or disagreements among the reviewers, they were resolved through discussion involving reviewers (TG and GGG). Before inclusion in the review, publications were independently assessed by three reviewers (WA, AH, and THA).

## Eligibility criteria

**Inclusion criteria.** Design: This study included all types of observational study designs.

Publication type: Both published and unpublished articles were considered.

Population: All parents whose neonates were admitted to the NICU in Ethiopia.

Intervention: Care provided at neonatal intensive care units.

Comparison: Not applicable.

Outcome: Parental satisfaction towards care given at NICUs and associated factors.

Study Setting: Only studies conducted in Ethiopia and based in institutional settings were included.

Language: Since there were no articles written in other language, only English articles were considered.

Publication year: This study included all published articles from June 2017 to December 30, 2023.

**Exclusion criteria.** This study excluded studies without full text and incomplete data on parental satisfaction and its associated factors. Additionally, studies with study design such as case reports, case series, and commentaries were also excluded from the analysis.

## Data extraction process and quality assessment

Five reviewers (TGH, TM, AB, NA, and GT) independently assessed and extracted data from the articles, considering overall study quality and suitability for inclusion in the review. The data extraction form for this study was created using a Microsoft Excel spreadsheet. A standardized data extraction format was used, which included information such as first author, study year, study design, study area, sample size, response rate, level of parental satisfaction, and odds ratio of factors influencing parental satisfaction with NICU services (S2 and S3 Tables).

The quality of the included studies was evaluated using the Newcastle-Ottawa Quality Assessment Scale (NOQAS) [29]. Each study was assessed based on criteria such as representativeness of the study, adequate sample size, acceptable non-response rate, use of a validated measurement tool, comparability of the study, description of outcome assessment, and use of appropriate statistical tests. A global rating score of seven or higher out of 10 was considered indicative of high quality [30] (S4 Table).

## Heterogeneity and publication bias

The $I^2$ statistic and Cochrane's $Q$ test was employed to evaluate the statistical heterogeneity among the included articles. The heterogeneity was classified as low, medium, or high heterogeneity based on the $I^2$ values of 25%, 50%, and 75%, respectively [31]. As there was heterogeneity observed among the included studies, the authors utilized a meta-analysis of random effects to estimate the aggregate pooled prevalence of parental satisfaction towards care given at NICU in Ethiopia.

For each study included in the analysis, the risk of bias was assessed independently by two authors (THA and TM). The Hoy 2012 tool, which comprises 10 recommended criteria for assessing the risk of bias in prevalence-based articles, was used for this purpose [32]. The criteria evaluated include representation of the population, sampling, methods of participant selection, non-response bias, data collection, acceptability of case definition, reliability and validity of study tools, methods of data collection, prevalence period, and appropriateness of the numerator and denominator. Each criterion was classified as having a low or high risk of bias, while an unclear assessment was classified as a high risk of bias. The overall risk of bias for each study was categorized based on the number of items with a high risk of bias: low ($\leq 2$), moderate (3–4), and high ($\geq 5$) (S5 Table). Additionally, Methods such as the funnel plot and Egger regression test were employed to assess publication bias. A significant p-value of 0.05 or less suggests the presence of publication bias [33].

## Statistical analysis

The data extracted from Microsoft Excel was exported to STATA version 14.0 software for further analysis. A random-effects model was utilized to estimate the pooled prevalence of parental satisfaction towards care given at NICU in Ethiopia. Meta-regression and subgroup analysis were conducted to identify potential sources of heterogeneity. The selected predictor variables' effects were analyzed using meta-analysis. The 'trim and fill' method, developed by Duval and Tweedie, was employed to estimate the approximate number of missing studies in the meta-analysis [34]. The results of the meta-analysis were presented using tables, a forest plot, and odds ratios (OR) with their corresponding 95% confidence intervals (CI). Additionally, to determine if there is any influence on the overall pooled prevalence of parental satisfaction with NICU services in Ethiopia, sensitivity analysis was performed by removing one study at a time. Additionally, by using the GRADE (Grades of Recommendation, Assessment, Development, and Evaluation) assessment tool to systematically assess the certainty of the evidence in our meta-analysis, we have assessed the quality of the evidence and developed recommendations based on that evidence [35] (S6 Table).

## Data management

To ensure consistency and adherence to the predetermined inclusion and exclusion criteria, a framework was developed to guide the screening and selection procedures. Prior to initiating the data extraction process, the framework/tool was tested and updated as needed. The search results obtained were then imported into EndNote X8 software to identify and remove any duplicate articles. This step is essential in ensuring that each unique article is considered for further evaluation and analysis.

### Data items

The first author, study year, study region, study design, sample size, level of parental satisfaction, odds ratio of factors affecting parental satisfaction with NICU services, and response rate were all included in the data extraction.

### Outcomes

The primary outcome of interest was the prevalence of parental satisfaction towards care given at NICU and its associated factors in Ethiopia.

## Results

### Articles included in this meta-analysis

In this study, the researchers initially identified a total of 1,264 articles through their search strategy from all databases. Duplicate articles, amounting to 512 were removed. Following the removal of duplicates, the remaining 752 articles underwent a title and abstract screening process. During this screening process, 394 articles were excluded as they did not meet the inclusion criteria based on their titles and abstracts alone. After the title and abstract screening, the researchers were left with 358 potentially relevant articles. These articles were then assessed in full-text form, meticulously evaluating them against the pre-defined selection criteria. Out of the 358 full-text articles, 347 were excluded based on specific reasons such as variations in study population or locations, did not focus on parental satisfaction with NICU services, lacking the reporting of the desired outcomes (S7 Table).

Finally, after this rigorous selection process, eleven studies met the inclusion criteria and were considered suitable for the final meta-analysis based on the pre-defined criteria and quality assessment. A PRISMA flow chart of the study selection shows the specific steps of the screening procedure (Fig 1).

### Description of the study

In this study, a total of 3,248 parents from 11 articles were included. The individual studies had sample size ranging from 109 to 422 participants, with response rates ranging from 93.5% to 100%. Majority of the selected articles utilized a cross-sectional study design. All the selected studies were conducted between 2017 and 2023. The study with the highest prevalence of parental satisfaction (77%) was conducted in the Amhara region [36], while the lowest prevalence (41.8%) was observed in a study conducted in Addis Ababa [37]. The eligible studies were distributed across five geographical regions, with four studies conducted in Amhara [36, 38–40], two studies in the Oromia region [41, 42], two studies in Addis Ababa [37, 43], two studies in the Harar region [44, 45], and one study in the Southern Nations, Nationalities, and Peoples' Region (SNNPR) [46].

The quality assessment using the Newcastle-Ottawa Scale indicated that there was no appreciable risk, and therefore, all the studies were considered in the systematic review and meta-analysis. For a more detailed overview of the characteristics of the included articles, please refer to Table 1.

### The estimated pooled prevalence of parental satisfaction

For this Mata analysis, we included eleven studies. The estimated pooled prevalence of parental satisfaction towards care given at NICU was 55% (95% CI: 49%–60%), but the random-effects model analysis revealed a significant high level of heterogeneity among the studies ($I^2$ = 90.45%, p = 0.00) (Fig 2).

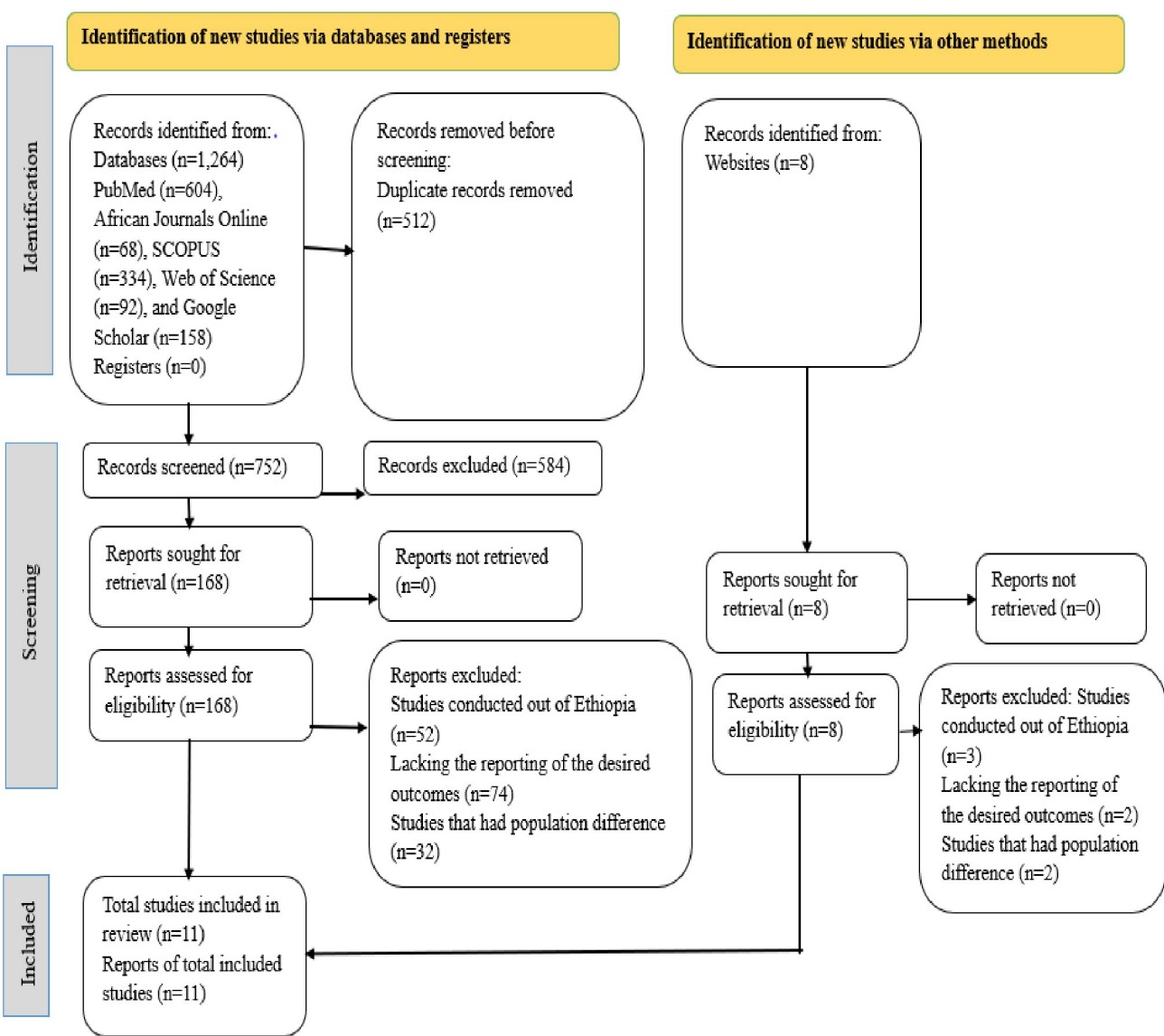

**Fig 1. PRISMA 2020 flow diagram of included studies in this systematic review and meta-analysis of parental satisfaction in Ethiopia.**

## Publication bias

The effect sizes of all the studies are regularly distributed around a funnel plot's center if there is no publication bias. As a result, the distribution of the result shown in (Fig 3) appears to be symmetrical in the funnel plot. Furthermore, there was no significant difference in the Egger's test results for the small study effect on publication bias. This suggests that when evaluating the pooled prevalence of parental satisfaction towards care given at NICU, there was neither publication bias nor small study impact among the included studies. In order to lessen and address publication bias in the studies, a trim-and-fill analysis was conducted for the prevalence of parental satisfaction towards care given at NICU. The outcome demonstrated that no study was imputed for missing studies, and the results estimated pooled prevalence and unadjusted prevalence were roughly equal.

**Table 1. Characteristics of studies included in the systematic review and meta-analysis of the prevalence of parental satisfaction towards care given at NICU in Ethiopia.**

| Authors | Study Year | Study Area | Study Design | Sample Size | Case | Prevalence (%) | Response Rate (%) | Quality score based on NOS |
|---|---|---|---|---|---|---|---|---|
| Fikadu L. et al. [41] | 2020 | Oromia | Mixed study | 109 | 79 | 72.75 | 100 | 8 |
| Jamie AH et al. [44] | 2023 | Harar | cross-sectional | 299 | 165 | 57 | 96.3 | 9 |
| Workie M et al. [45] | 2023 | Harar | cross-sectional | 408 | 206 | 50.5 | 97.6 | 9 |
| Berhan Y. [43] | 2020 | Addis Ababa | cross-sectional | 299 | 128 | 42.8 | 95.7 | 8 |
| Ali MS. et al. [38] | 2020 | Amhara | cross-sectional | 317 | 150 | 50 | 94.6 | 9 |
| Alle YF et al. [39] | 2022 | Amhara | cross-sectional | 385 | | 47.8 | 95.06 | 9 |
| Alemu A. et al. [40] | 2021 | Amhara | cross-sectional | 400 | 214 | 55 | 97.3 | 8 |
| Endale H. [37] | 2017 | Addis Ababa | cross-sectional | 422 | 167 | 41.8 | 94.7 | 8 |
| Sileshi E et al. [46] | 2022 | SNNPR | cross-sectional | 401 | 240 | 63 | 95 | 9 |
| Adal Z et al. [42] | 2021 | Oromia | cross-sectional | 122 | 66 | 57.9 | 93.5 | 7 |
| Mekonnen WN et al. [36] | 2017 | Amhara | cross-sectional | 129 | 98 | 77 | 98.4 | 7 |

NOS: Newcastle Ottawa Scale

SNNPR: Southern Nations, Nationalities, and Peoples' Region

## Subgroup analysis

We have performed a subgroup meta-analysis according to the year and study region. The goal was to assess if the grouping under consideration aids in explaining some of the observed between-study heterogeneity by comparing the pooled estimates among the groups. Consequently, the pooled prevalence estimates in Oromia were slightly higher, 63% (95% CI: 57%–69%). However, as can be seen in the forest plots (Fig 4) by region, the subgroup analysis did not alter the observed heterogeneity ($I^2$ = 90.45%, p≤0.000) and the subgroup analysis by year also indicated the same result ($I^2$ = 90.45%, p≤0.000) (Fig 5). To determine if the sample size and study year were sources of heterogeneity for the pooled prevalence, meta regression was performed with these variables considered, but none of them was also significant.

## Sensitivity analysis

In order to determine if there was any influence on the overall pooled prevalence of parental satisfaction with NICU services in Ethiopia, sensitivity analysis was performed by removing one study at a time. The results indicated that no study had a significant impact on the overall pooled estimated prevalence. Consequently, as shown in Fig 6, the 95% CI varied from (0.48, 0.59) to (0.47, 0.61) when the named specific study was excluded.

## Factors associated with parental satisfaction with NICU services in Ethiopia

The pooled effect estimates of various factors associated with parental satisfaction with NICU services in Ethiopia were significant predictors, showing a significant association with birth weight (AOR = 1.59, 95% CI: 0.86–2.95), length of hospital stays (AOR = 3.44, 95% CI: 2.17–5.44), place of residence (AOR = 3.16, 95% CI: 1.77–5.63), and availability of necessary information using direction indicator (AOR = 3.39, 95% CI: 2.09–5.51), with a significant level of heterogeneity ($I^2$ = 44.6%, p≤0.179), ($I^2$ = 24.4%, p≤0.266), ($I^2$ = 76.4%, p≤0.005), and ($I^2$ = 0.0%, p≤0.466) respectively.

**Birth weight.** This meta-analysis included two studies [39, 44], and the result indicated a significant association between normal birth weight and parental satisfaction (Fig 7). Parents

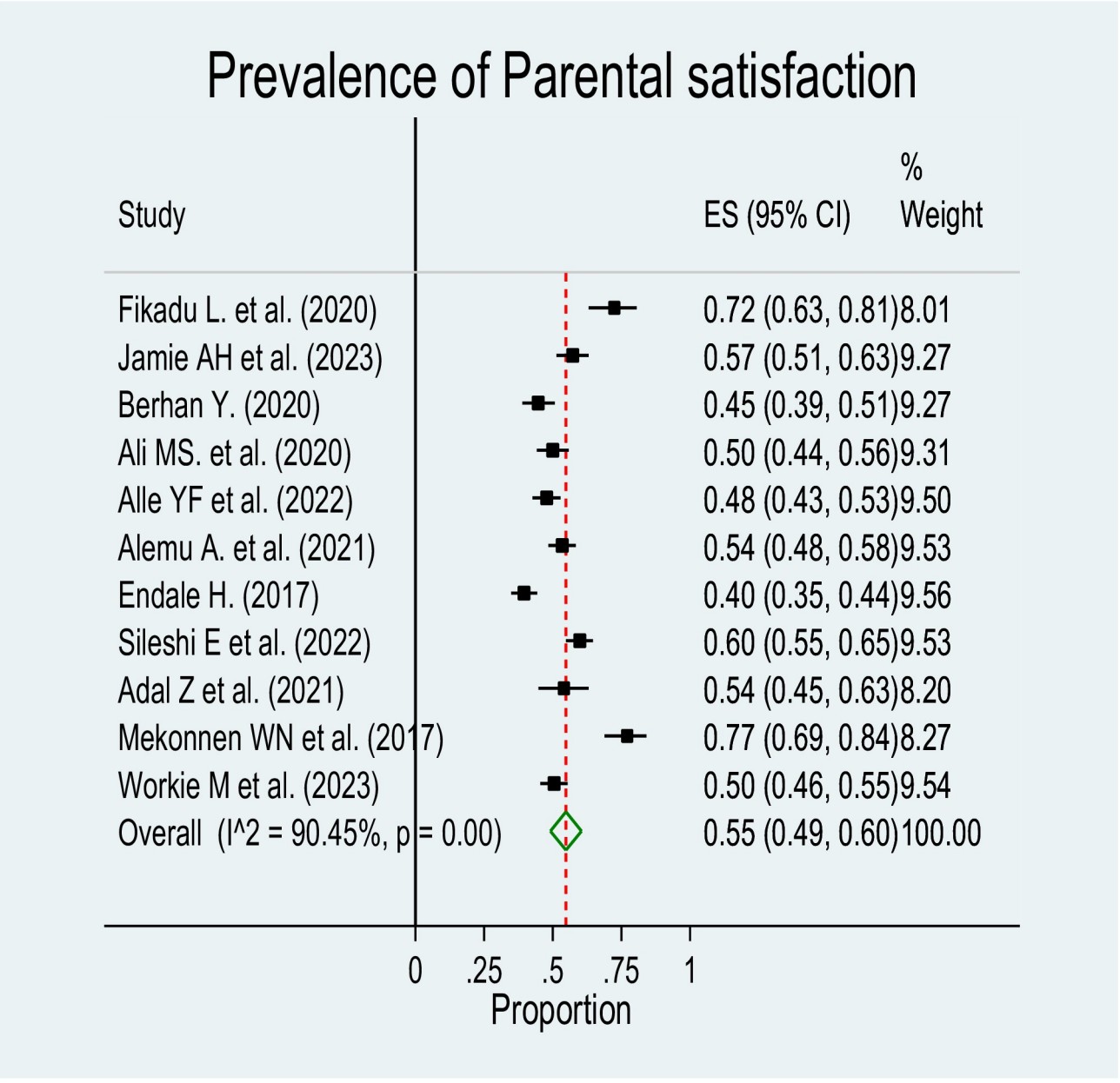

**Fig 2. Forest plot for the prevalence of parental satisfaction towards care given at NICU in Ethiopia.**

who had baby with normal birth weight were approximately 2 times more likely satisfied with care given at NICU service compared to their counterpart.

**Length of hospital stays.** This meta-analysis included three studies [38–40, 45]. The pooled meta-regression analysis revealed a statistically significant correlation between length of hospital stays and parental satisfaction with NICU services (AOR = 3.44, 95% CI: 2.17–5.44) (Fig 8).

**Place of residence.** In this review, the relationship between place of residence and parental satisfaction with NICU services was examined. The findings from a meta-regression analysis, which incorporated data from four studies [39, 41, 43–45], demonstrated a positive

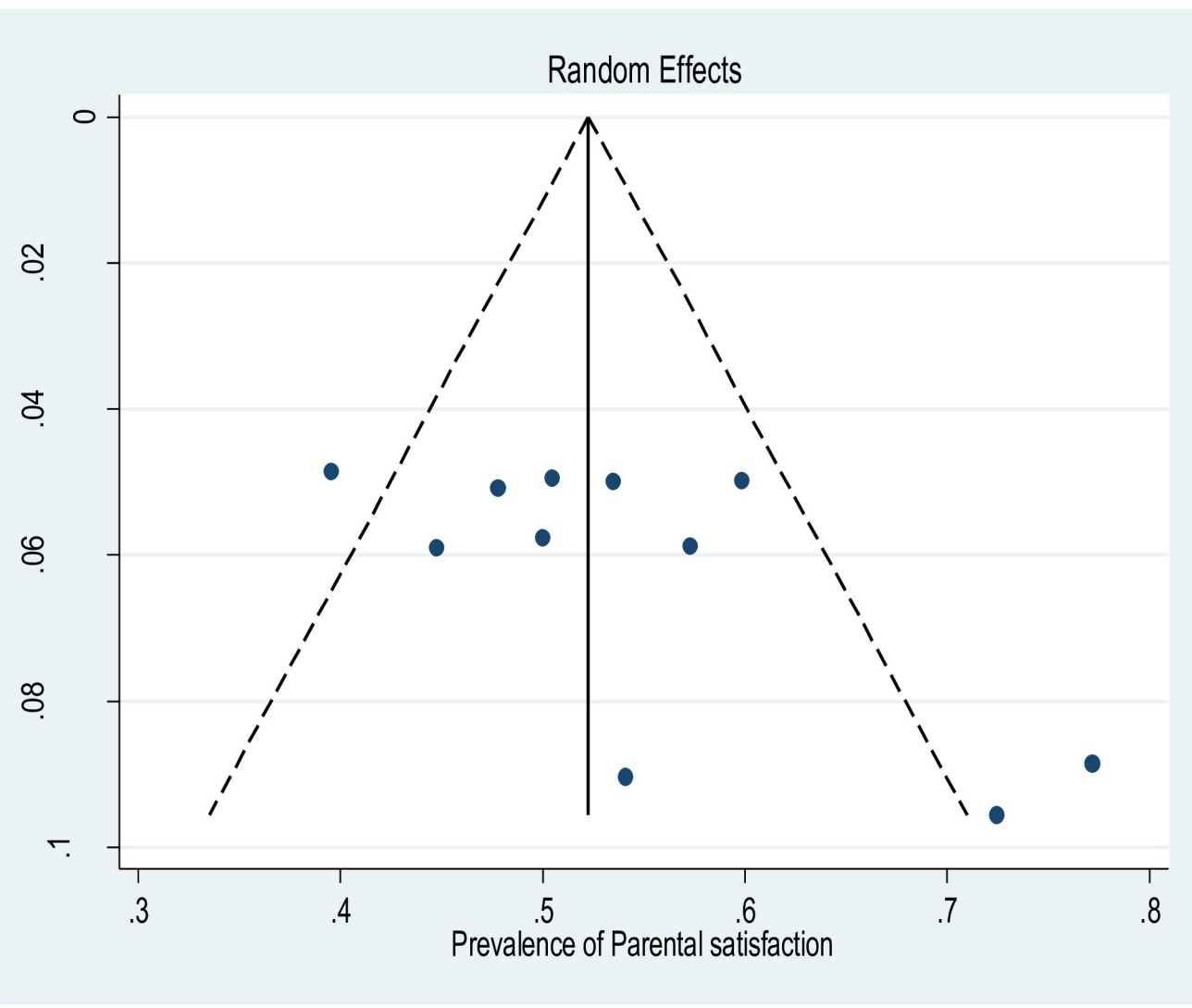

**Fig 3. Funnel plot included distribution of studies in parental satisfaction towards care given at NICU in Ethiopia.**

association between place of residence and parental satisfaction with NICU services (AOR = 3.16, 95% CI: 1.77–5.63) (Fig 9). Parents residing in rural areas were approximately three times more likely to be satisfied with the care provided at the NICU service compared to parents residing in urban areas.

**Availability of necessary information using direction indicator.** A significant association was discovered between the availability of necessary information using direction indicators in the hospital and parental satisfaction with NICU services (AOR = 3.39, 95% CI: 2.09–5.51), as depicted in Fig 10. The meta-analysis comprised two studies [36, 46]. Parents whose baby was admitted to the hospital and had access to necessary information through the use of direction indicators were approximately 3.39 times more likely to be satisfied with the care provided at the NICU service compared to parents who did not have this access.

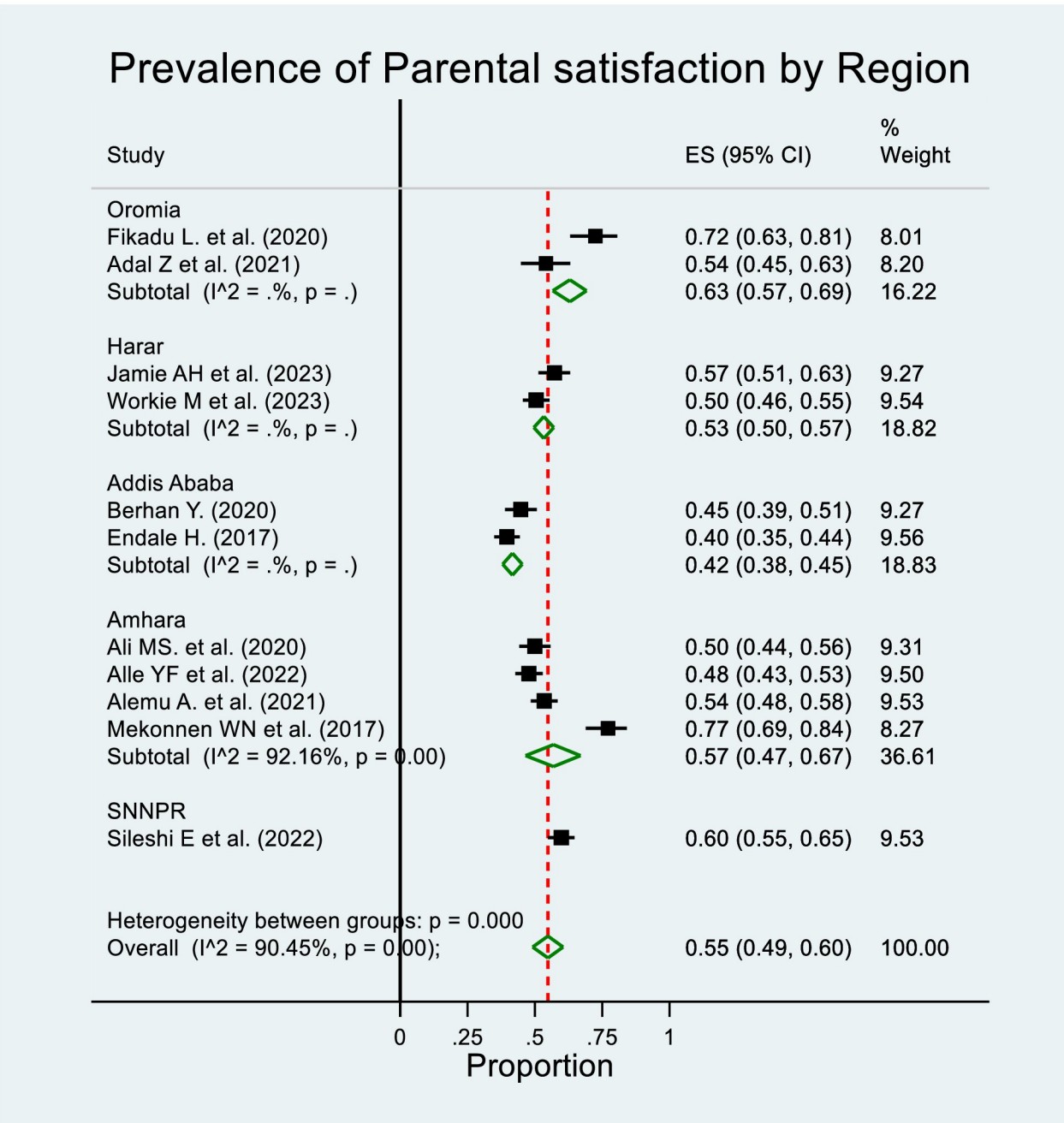

**Fig 4. Forest plot for the prevalence of parental satisfaction with NICU services by region.**

## Discussion

The purpose of this study is to find out how satisfied parents are with the care they receive in the NICU. For newborns with complicated medical issues or those born prematurely, the neonatal intensive care unit, offers critical care. Since the NICU setting can be emotionally challenging for parents, it is important to know how satisfied they are with the care they are receiving in order to make the experience better for both parents and their babies. The pooled

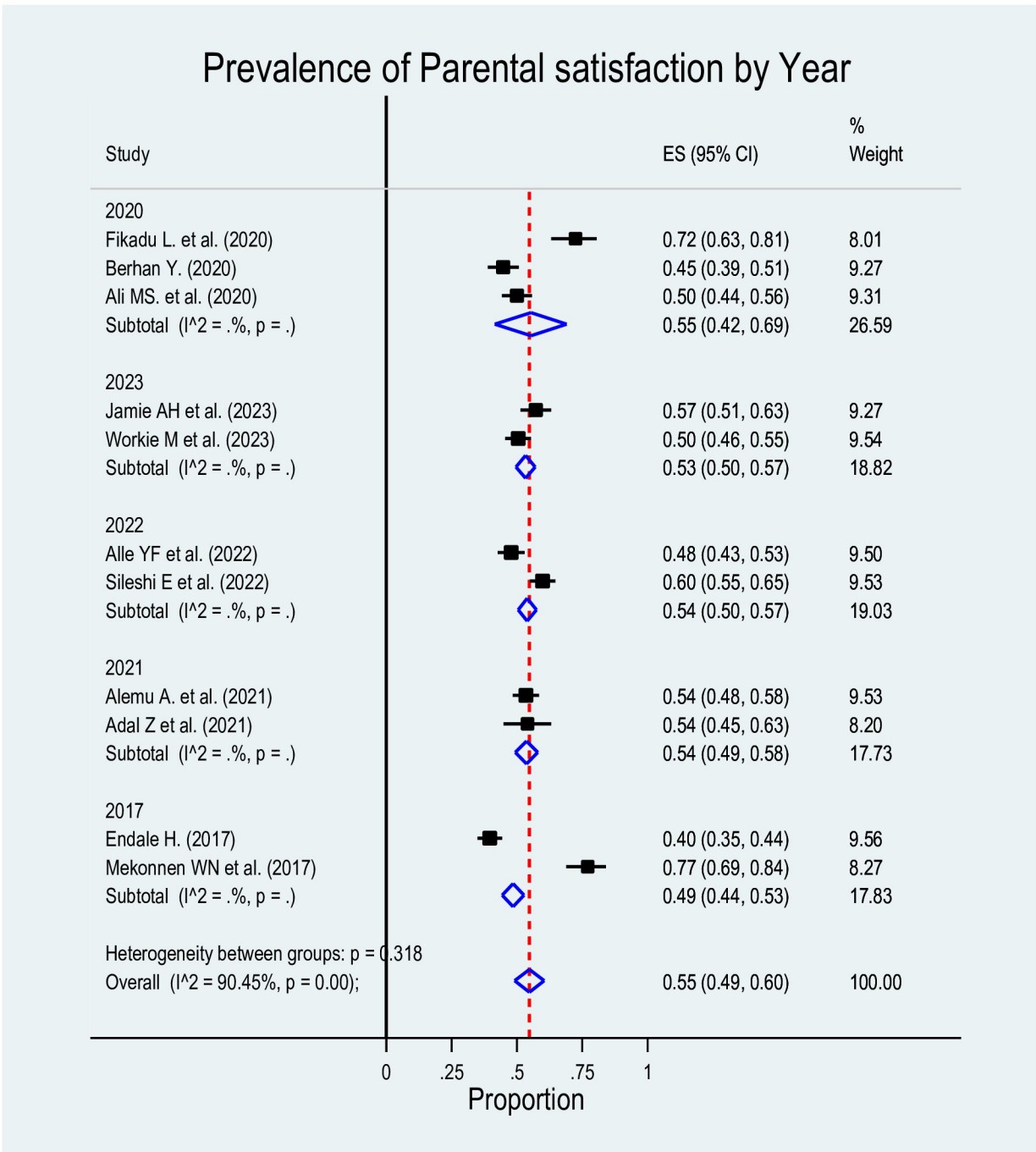

**Fig 5. Forest plot for the prevalence of parental satisfaction towards care given at NICU by year in Ethiopia.**

prevalence of parental satisfaction towards care given at NICU in Ethiopia, according to this study, is 55% (95% CI: 49%–60%).

The result of our study are significantly less than the studies carried out in the Rwanda, Kenya, and Sri Lanka with the same study design, which revealed 66%, 81%, 94.6% [47–49] respectively. Similarly, studies carried out in Iran, Norway, and Greek also showed that 63.6%,

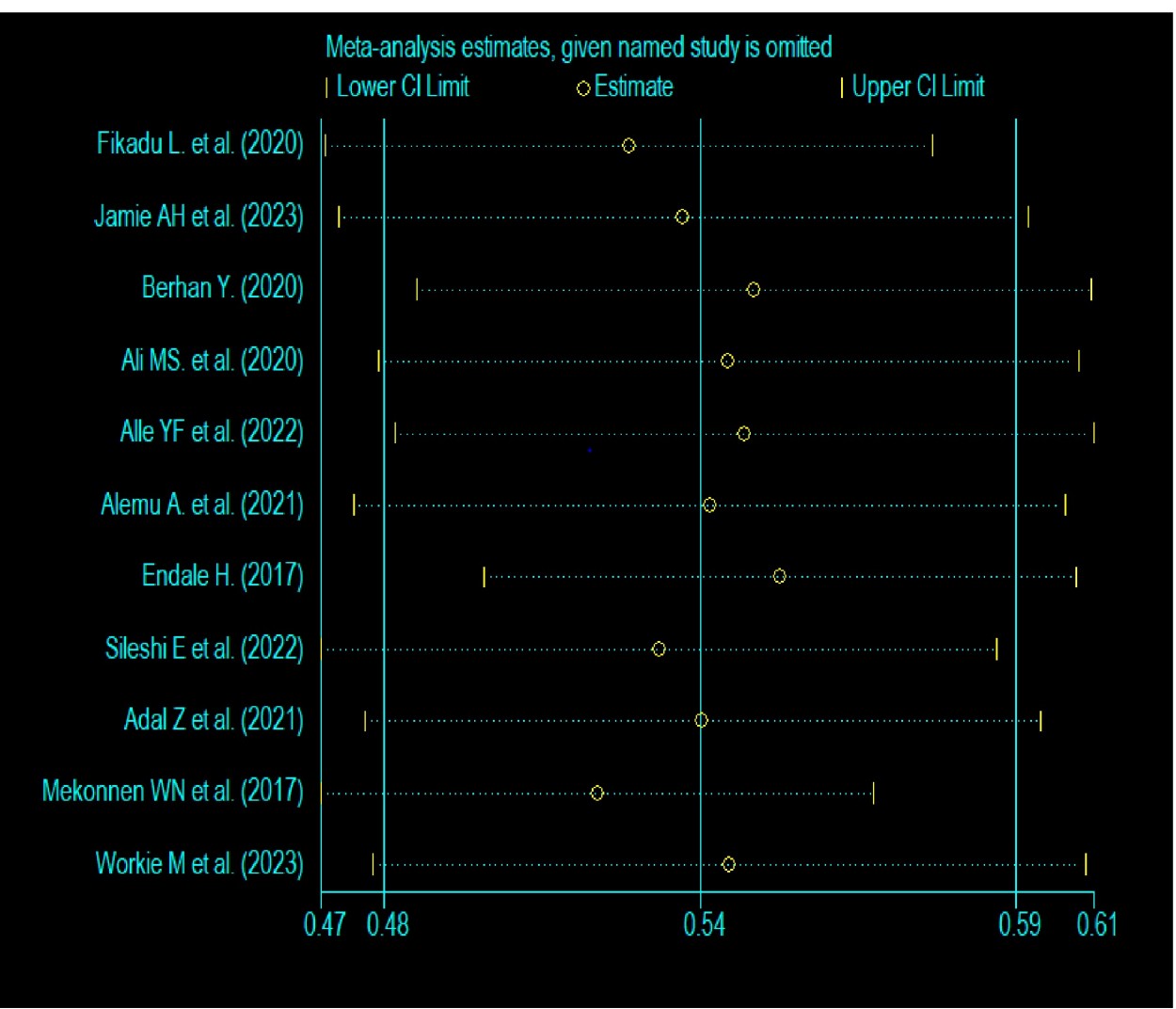

**Fig 6. Meta-analysis result when omitting the named particular study.**

65.7%, 92.6%, 87.8% [50–52] of them, respectively. The difference might be due to the difference in availability and quality of healthcare infrastructure and resources, variations in the level of medical technology, staffing ratios, and access to specialized care, effective communication between healthcare providers and parents, and cultural attitudes towards healthcare and medical interventions can impact parental satisfaction.

Our result (55%) is similar to the finding of a study conducted in Nepal in 2019 (56.7%) [53]. The previous study used a cross-sectional descriptive study design in neonatal care units of five selected public hospitals.

According to this study, parents who had a baby with a normal birth weight were roughly twice as likely as those who did not to be satisfied with the NICU's treatment. This result is consistent with previous research conducted in Kenya, California, and Nepal [48, 53, 54], which also showed a strong correlation between a normal birth weight and parental satisfaction with NICU care. Based on these results, it is critical that healthcare practitioners and NICU staff focus efforts to support infants in obtaining and maintaining a healthy birth

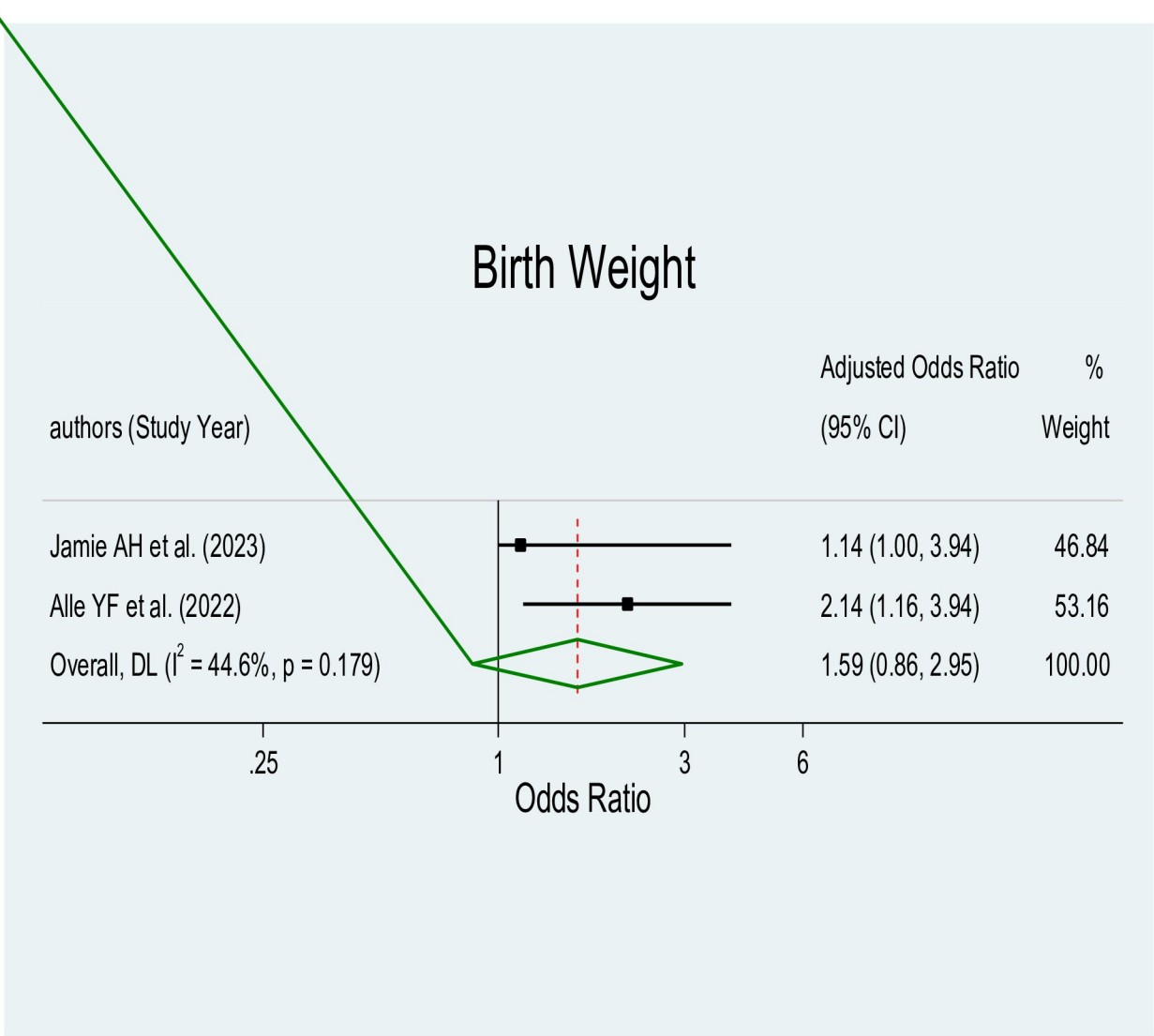

**Fig 7. Forest plot of birth weight for parental satisfaction with NICU services.**

weight. This may involve implementing interventions to promote optimal prenatal care, early detection and management of potential complications, and providing additional support to infants with low birth weights. Healthcare providers may be able to improve parent satisfaction with NICU care and improve the general health of infants and their families by concentrating on improving birth weight outcomes.

Our meta-regression analysis showed that parents whose babies had a shorter length of hospital stay in the neonatal intensive care unit were about three times more likely to report satisfaction with the care delivered compared to parents whose babies had longer hospital stays. This result is in line with research done from Iran, Vietnam, Norway, and Greece [26, 50, 51, 55]. The implication here is that shorter hospital stays for newborns in the NICU might improve parental satisfaction. It suggests that health care providers and NICU staff ought to try to maximize the care given to newborns, making sure that their hospital stay is as short as possible while still meeting their medical requirements. This may involve implementing

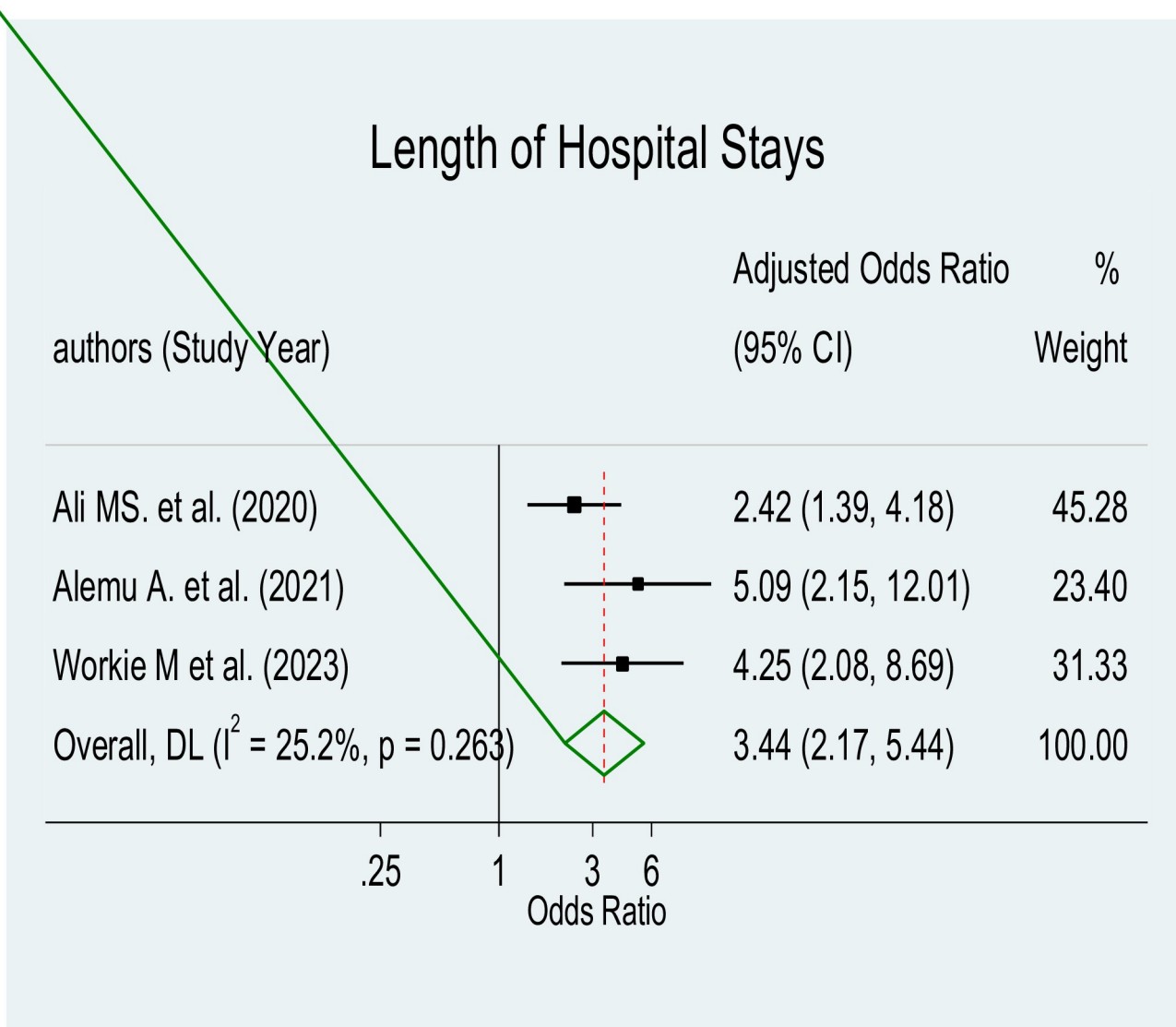

**Fig 8. Forest plot of length of hospital stay for parental satisfaction with NICU services.**

strategies to improve efficiency, enhance coordination among healthcare professionals, and provide appropriate support and care for infants to promote their timely recovery and discharge from the NICU.

According to this study, parents who lived in rural areas were around three times more likely than those who lived in urban areas to be satisfied with the treatment received by their neonates at the NICU. This result is in line with research conducted in Greece [55].

Our research revealed a relationship between parental satisfaction with NICU care and the availability of important information using a direction indicator. This result is in consistent with research conducted in Kenya, Sri Lanka, and London [48, 49, 56]. According to the results, parents are more likely to express satisfaction with the treatment they receive when they have access to crucial information via the usage of direction signs in the NICU. It also implies that healthcare facilities should prioritize providing clear and easily accessible

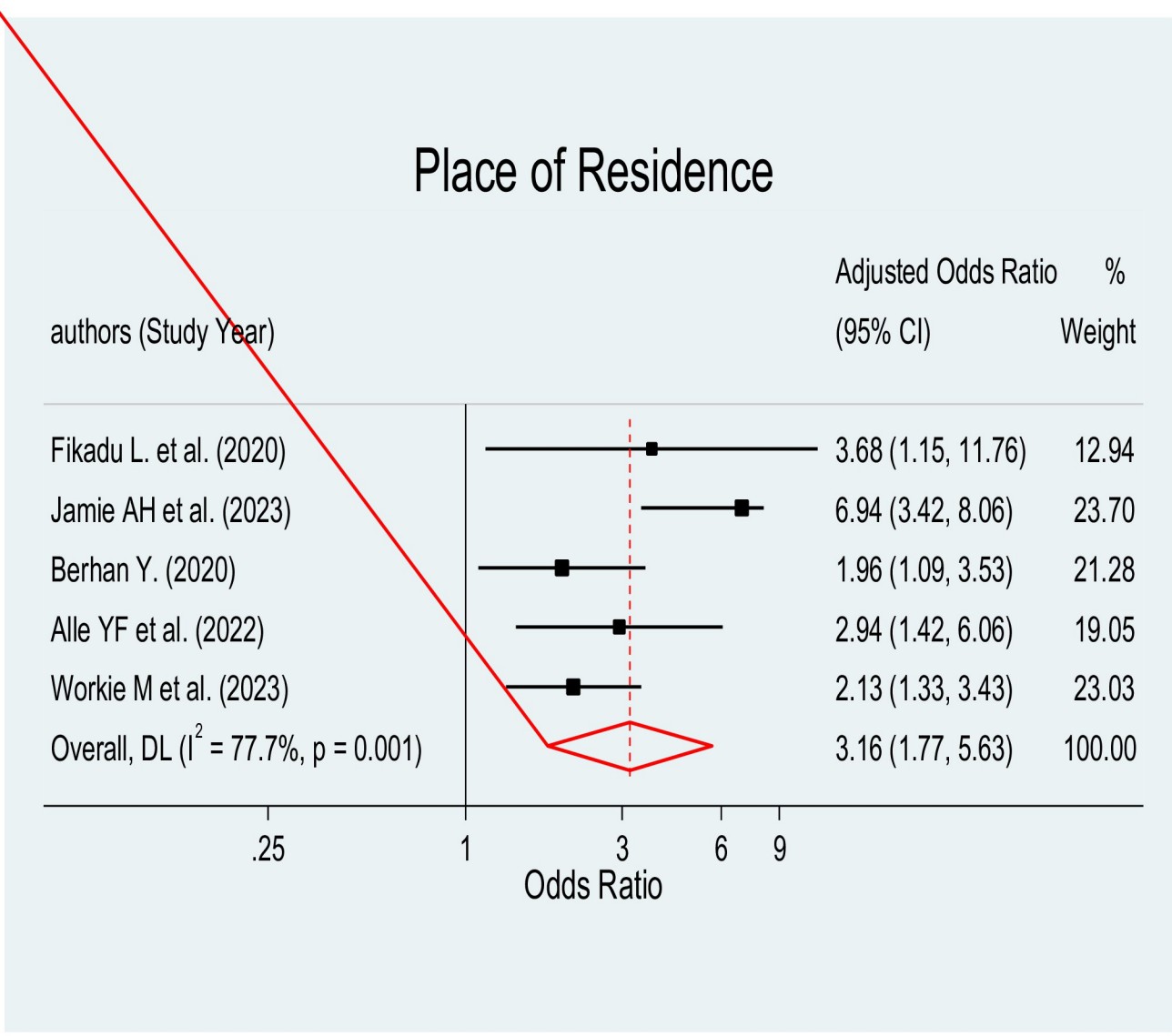

**Fig 9. Forest plot of place of residence for parental satisfaction towards care given at NICU in Ethiopia.**

information to parents, using tools such as direction indicators, to enhance their understanding and involvement in the care of their neonates in the NICU. This can contribute to better outcomes for both parents and neonates during their NICU stay.

## Strengths and limitations of this study

This review closely follows the 2020 PRISMA checklist, improving quality for the audience. In addition, this research will shed light on how to better support newborns and their families in NICUs, educating administrators and legislators as well as offering suggestions for future clinical practice and guidelines to raise the standard of care given their overall. This review's limitations include the potential for self-report bias resulting from the original research that was included, in addition to the previously indicated strengths. This study was limited to Ethiopia. To address this limitation, we recommended conducting similar studies in African countries.

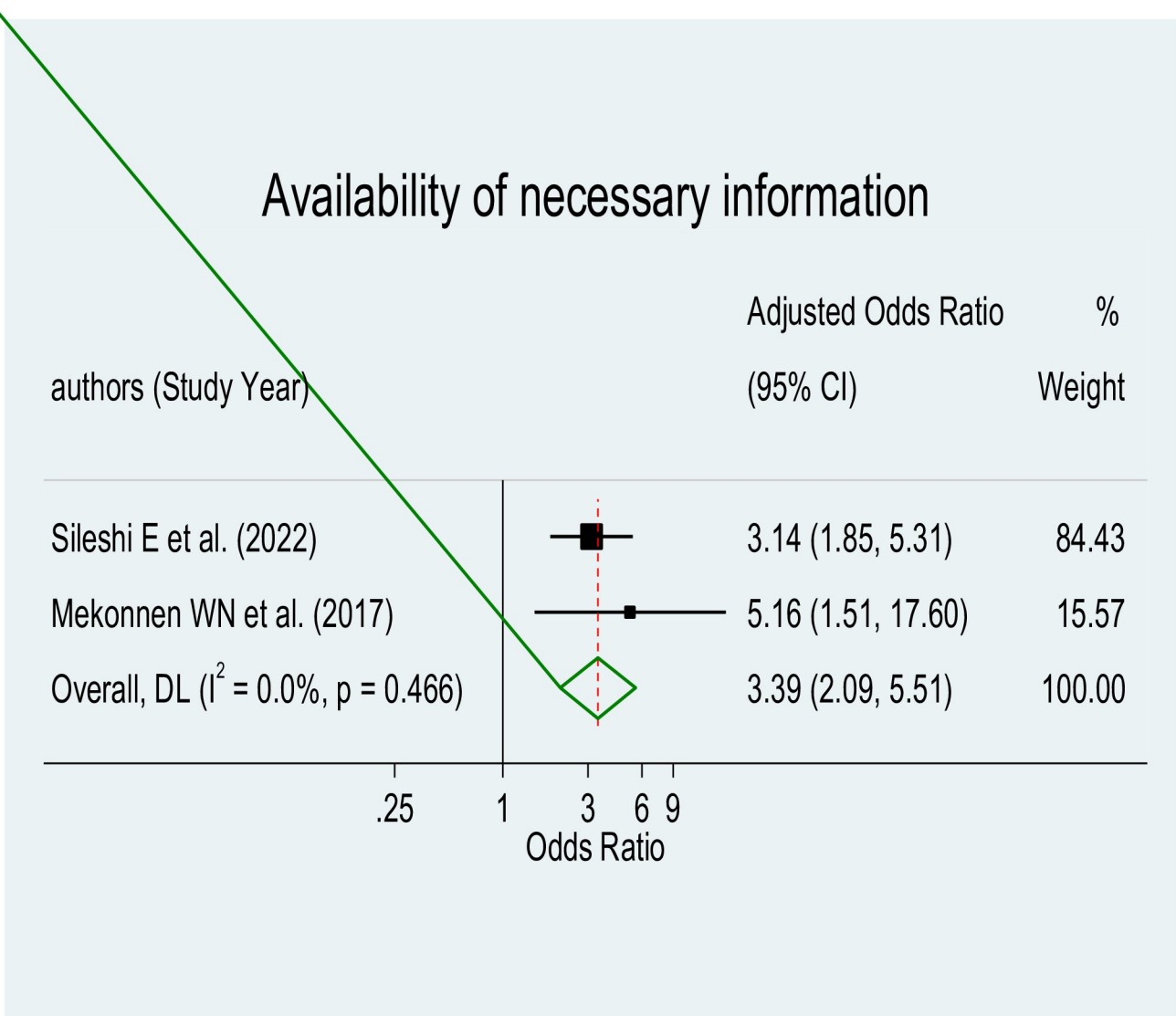

**Fig 10. Forest plot of availability of necessary information using direction indicators for parental satisfaction with NICU services in Ethiopia.**

## Conclusion

This study's pooled prevalence is 55%. Thus, the study's findings suggest that, compared to past research that were published in certain countries, the pooled prevalence of parental satisfaction with NICU care is lower. Parental satisfaction's lower pooled prevalence suggests a possible area for NICU service improvement.

Low birth weight, urban residences, lack of availability of important information using a direction indicator, and long length of hospital stay were contributing factors to the lower pooled prevalence of parental satisfaction. These factors may heighten parental stress and anxiety and raise questions about the newborn's long-term health. Living in an urban area may raise your expectations for medical treatment. Parental satisfaction may be lower if these expectations are not fulfilled. Insufficient knowledge about a baby's condition, course of treatment, and aftercare can cause parents to lose faith in medical professionals and experience a

decrease in parental satisfaction. Lower parental satisfaction can be caused by a long hospital stay, which leads to financial strain, employment obligations, and prolonged separations from other family members.

In order to improve the experience for both parents and their newborns, it is crucial to understand how satisfied parents are with the care their babies are receiving, as the NICU setting can be emotionally taxing for them. Efforts should be on the identified specific factors contributing to lower satisfaction and develop strategies to address them effectively. This could entail assessing and resolving any gaps in the care that may exist, making sure that parents are informed in a timely and correct manner, and fostering a compassionate and encouraging environment in the NICU. Healthcare facilities can improve the entire experience of both parents and infants, which can lead to improved outcomes and a more positive care environment, by working to raise the satisfaction levels of parents in the NICU.

## Supporting information

**S1 Table. PRISMA 2020 Checklist followed for this systematic review and meta-analysis.** (DOCX)

**S2 Table. Extracted studies which are eligible for systematic review and meta-analysis of parental satisfaction in Ethiopia.** (DOCX)

**S3 Table. Extracted studies for systematic review and meta-analysis of factors associated with parental satisfaction in Ethiopia.** (DOCX)

**S4 Table. Newcastle-Ottawa Quality Assessment Scale.** (DOCX)

**S5 Table. Risk of bias assessment of the included studies was conducted using the Hoy 2012 tool.** (XLSX)

**S6 Table. GRADE assessment to assess the certainty of evidence of this systematic review and meta-analysis of parental satisfaction in Ethiopia.** (DOCX)

**S7 Table. Studies identified during the literature search for systematic review and meta-analysis of parental satisfaction in Ethiopia.** (DOCX)

## Acknowledgments

We express our gratitude to the authors of the included papers for their valuable contributions to this meta-analysis and systematic review.

## Author Contributions

**Conceptualization:** Teklehaimanot Gereziher Haile, Gebreamlak Gebremedhn Gebremeskel, Teklewoini Mariye, Abrha Hailay, Teklehaymanot Huluf Abraha, Girmay Teklay.

**Data curation:** Teklehaimanot Gereziher Haile, Gebreamlak Gebremedhn Gebremeskel, Teklewoini Mariye, Guesh Mebrahtom, Woldu Aberhe, Abrha Hailay, Teklehaymanot Huluf Abraha, Negasi Asres, Teklay Guesh, Girmay Teklay.

**Formal analysis:** Teklehaimanot Gereziher Haile, Gebreamlak Gebremedhn Gebremeskel, Teklewoini Mariye, Woldu Aberhe, Abrha Hailay, Negasi Asres.

**Investigation:** Teklehaimanot Gereziher Haile, Gebreamlak Gebremedhn Gebremeskel, Teklewoini Mariye, Guesh Mebrahtom, Woldu Aberhe, Abrha Hailay, Teklehaymanot Huluf Abraha, Negasi Asres, Teklay Guesh, Girmay Teklay.

**Methodology:** Teklehaimanot Gereziher Haile, Gebreamlak Gebremedhn Gebremeskel, Teklewoini Mariye, Guesh Mebrahtom, Woldu Aberhe, Abrha Hailay, Teklehaymanot Huluf Abraha, Negasi Asres, Teklay Guesh, Girmay Teklay.

**Project administration:** Teklehaimanot Gereziher Haile, Gebreamlak Gebremedhn Gebremeskel, Teklewoini Mariye, Guesh Mebrahtom, Woldu Aberhe, Abrha Hailay, Teklehaymanot Huluf Abraha, Negasi Asres, Teklay Guesh, Girmay Teklay.

**Resources:** Teklehaimanot Gereziher Haile, Gebreamlak Gebremedhn Gebremeskel, Teklewoini Mariye, Guesh Mebrahtom, Woldu Aberhe, Abrha Hailay, Teklehaymanot Huluf Abraha, Negasi Asres, Teklay Guesh, Girmay Teklay.

**Software:** Teklehaimanot Gereziher Haile, Gebreamlak Gebremedhn Gebremeskel, Teklewoini Mariye, Woldu Aberhe, Teklehaymanot Huluf Abraha, Negasi Asres, Teklay Guesh, Girmay Teklay.

**Supervision:** Teklehaimanot Gereziher Haile, Gebreamlak Gebremedhn Gebremeskel, Teklewoini Mariye, Guesh Mebrahtom, Woldu Aberhe, Abrha Hailay, Teklehaymanot Huluf Abraha, Negasi Asres, Teklay Guesh, Girmay Teklay.

**Validation:** Teklehaimanot Gereziher Haile, Gebreamlak Gebremedhn Gebremeskel, Teklewoini Mariye, Guesh Mebrahtom, Woldu Aberhe, Abrha Hailay, Teklehaymanot Huluf Abraha, Negasi Asres, Teklay Guesh, Girmay Teklay.

**Visualization:** Teklehaimanot Gereziher Haile, Gebreamlak Gebremedhn Gebremeskel, Teklewoini Mariye, Guesh Mebrahtom, Woldu Aberhe, Abrha Hailay, Teklehaymanot Huluf Abraha, Negasi Asres, Teklay Guesh, Girmay Teklay.

**Writing – original draft:** Teklehaimanot Gereziher Haile.

**Writing – review & editing:** Teklehaimanot Gereziher Haile, Gebreamlak Gebremedhn Gebremeskel, Teklewoini Mariye, Guesh Mebrahtom, Woldu Aberhe, Abrha Hailay, Teklehaymanot Huluf Abraha, Negasi Asres, Teklay Guesh, Girmay Teklay.

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
