## [Decision Letter · Decision Letter 0]

18 Jul 2024

PONE-D-24-18838Parental satisfaction towards care given at neonatal intensive care unit in Ethiopia: a systematic review and meta-analysisPLOS ONE

Dear Dr. Haile,

Thank you for submitting your manuscript to PLOS ONE. After careful consideration, we feel that it has merit but does not fully meet PLOS ONE’s publication criteria as it currently stands. Therefore, we invite you to submit a revised version of the manuscript that addresses the points raised during the review process. 

Find my suggestions here below:

    Indicate the PROSPERO Registration Number:

        Clearly state the PROSPERO registration number for this systematic review and meta-analysis (SRMA) in the manuscript.

    Use Appropriate Regional Nomenclature:

        Ensure that the appropriate and correct naming of regions within the respective country is consistently used throughout the manuscript. This includes all text references, figures (such as Figure 4), tables, and any supplementary materials. for example Oromo Vs. Oromia, Harar Vs. Harari, Southern Vs. SNNPR etc.

    Detailed Results Section in Abstract is needed:

        Expand the results section of the abstract to provide more detailed and engaging information. Include key findings, statistical significance, and any important trends or patterns observed in the data. This will make the abstract more informative and appealing to the audience.

    Change Subtopic Title:

        Change the title of the subtopic from "Result" to "Results" to reflect the content accurately.

We look forward to receiving your revised manuscript.

Kind regards,

Tamirat Getachew

Academic Editor

PLOS ONE

Journal Requirements:

2. In the online submission form, you indicated that [This study was not submitted to any journal for publication. All raw data generated or analyzed during the current investigation will be made available upon request by the corresponding author.].

Reviewers' comments:

Reviewer's Responses to Questions

**Comments to the Author**

1. Is the manuscript technically sound, and do the data support the conclusions?

Reviewer #1: Partly

Reviewer #2: Yes

Reviewer #3: Yes

Reviewer #4: Yes

2. Has the statistical analysis been performed appropriately and rigorously? 

Reviewer #1: Yes

Reviewer #2: Yes

Reviewer #3: Yes

Reviewer #4: Yes

3. Have the authors made all data underlying the findings in their manuscript fully available?

Reviewer #1: No

Reviewer #2: Yes

Reviewer #3: Yes

Reviewer #4: Yes

4. Is the manuscript presented in an intelligible fashion and written in standard English?

Reviewer #1: Yes

Reviewer #2: Yes

Reviewer #3: Yes

Reviewer #4: Yes

5. Review Comments to the Author

Reviewer #1: Thanks to the authors of the manuscript for trying to synthesize evidence on patient satisfaction in NICU. In my review, I have found some methodological issues and I have organized my feedback chronological to the chapters and subchapters of this manuscript. Please find the followings:

1) Abstracts

1.1: In the abstract's method section, if we have any word limitation then, I would prefer mentioning about the risk of bias of this SRMA, rather than mentioning which software we use to extract data or analyse the data.

2) Introduction

2.1: It is ambitious to say, this the first kind of study, I'd prefer rephrasing this sentence as this study might be the extension of the available evidence. Also, I just found one published SRMA on the similar topic from Ethiopia (https://www.ncbi.nlm.nih.gov/pmc/articles/PMC11000344/).

3) Methods

3.1: Language Restriction: Including only English articles in the era of internet, AI, google translators, proves your literature search might miss some of the important articles published in other language.

3.2: Exclusion criteria: Excluding articles that did not report on the prevalence of parental satisfaction is an inappropriate approach for this type of SRMA where the main focus is on patient satisfaction. Most of the studies use different scales to report patient satisfaction and they have different levels of reporting. Even for this particular research, all the included studies did not use the same tool to assess the patient satisfaction. I would suggest the authors to read Cochrane SRMA handbook and their published SRMA to have a clear idea how to report SRMAs with different scales and different reporting system.

3.3: Exclusion Criteria: How did the authors define, methodological problems, such as inadequate sample size, sampling techniques, incomplete data, inadequate statistical analysis, or any other relevant methodological limitations? What are their thresholds of exclusion? This needs to be mentioned in the Method section clearly.

3.4: The Holy 2012 tool can be used to assess the Risk of bias of the prevalence based articles, it doest not assess the internal and external validity of the articles. Please omit or rephrase the sentence.

3.5: Why the authors assess Risk of Bias using two different scales (NOQAS and Holy tool) since authors only included the prevalence studies for this SRMA?

3.6: Statistical analysis: Authors are reporting prevalence of satisfaction, it is unclear why authors have done the meta-regression here.

3.7: GRADE assessment is an important tool to assess the certainty of evidence of your meta-analysis, which authors skipped. I'd suggest add this to the manuscript.

4) Results

4.1: What did the authors do since they got very high heterogeneity? They did mention they explored their heterogeneity through subgroup analysis. However, in their forest plots, i did not find and I^2 value for the subgroups, which is very perplexing for the authors. They did report I^2 value in their "Results: Subgroup Analysis" section; however, I believe authors have interpreted the heterogeneity in their subgroup analysis incorrectly. The I^2 = 90.45% is the overall heterogeneity. Please follow the Cochrane SRMA handbook or other available literatures.

4.2: For the forest plots, please try to add the I^2 values for each of the subgroups, chi-square statistics, and p-value.

Reviewer #2: Review Reports

Title: Parental satisfaction towards care given at neonatal intensive care unit in Ethiopia: a systematic review and meta-analysis

Manuscript Number: PONE-D-24-18838

Comments

• The issues is contemporary agenda.

• The background fails to entail all what it intends to entail. Avoid repeated use of sentences. E.g. Adverse consequences of dis-satisfaction.

• The methods fail to describe whether the inclusion of unpublished articles and its where about.

• The extractors, ambiguity an dhow the ambiguity was solved as well as the inclusion of retracted publication is not mentioned.

• The PCO is not described

• The result and the discussion should be brief and clear.

• The discussion should be relevant contextual and explanatory

• Edit for language and grammar E.g. Use ‘Oromia’ instead of Oromo.

Regards,

Reviewer #3: Parental satisfaction towards care given at neonatal intensive care unit in Ethiopia: a systematic review and meta-analysis

Reviewer Comment:

Thank you. I am pleased to review the following manuscript. I hope incorporating the following comment would enhance the quality of the manuscript at all.

Thank you.

Abstract:

1. The method section should contain the timeframe of relevant published articles.

2. The result should reflect the aim thus some important relevant associated factors should be present.

3. There is some inconsistency or may the author could not reflect which tried to express in the conclusion section 42-44 line. Try to simplify the line.

4. PROSPERO Reg. number is missing. Without this Systematic review is incomplete.

Introduction:

1. Try to rephrase the 47 line its just a copy of abstract.

2. According to line 50, WHO reference is not appropriately given.

3. Although the introduction section is well written but I think it has missing some important factors like, defining neonatal period, describing the existing NICU care given prevalence in other countries. Being a review paper, it should explain how patient satisfaction is important to reduce IMR using data.

Methods and materials:

1. The search term should contain “father’s satisfaction”, “Ethiopia” should be specified using breakdown of southern part or northern part.

2. In the study selection process, three researchers review but what about the disagreements was how resolved? It is not described at all. What about the Full text screening process?

3. Line 125 how unpublished articles could be incorporated in a systematic review? It will reduce the reliability of the systematic review.

4. The exclusion criteria should be more specific regarding the caregivers, timeframe, only government facility or private NICU facility, study type (case report, case series, review papers etc.). This factors should be described in a well written manner.

5. The PICO framework or CoCoPop format should be followed.

6. Line 144-145, discrepancies was resolved by TG and GGG, but was it in full text screening phase or initial title-abstract screening phase? Which software was used to screen the initial phase should be mentioned.

Results:

1. The result section should add some demographic information of the parents. Preferably a Table.

2. Line 288-289 should mention the exact AOR regardless three times.

Discussion:

1. The discussion should contain comparison with other region also regardless Nepal with representing the data.

2. The compassion and sub-group analysis should distinguish showing the OR. How one group is more satisfied with another with the respective OR. Ref (Parental satisfaction with neonatal intensive care unit services and associated factors in Ethiopia: systematic review and meta-analysis. DOI: https://doi.org/10.1186/s12912-024-01902-3).

3. Line 345-346, this line is contradictory. I think, this similar type of review was done in the previously mentioned DOI. I have conflict of interest with this line.

4. Regardless of Africa only choosing Ethiopia is one of the greatest limitation of the study I think.

5. How associated factors contribute to this study should precisely mention in the conclusion section also.

6. Authors contribution section should mention detailing the screening and publication bias section aslo.

Reference:

1. Ref 3, 30, 36, 38, 40, 45 should be rechecked and followed appropriate guideline.

Others:

1. Fig 1 should mention PRISMA, variation of study means (location)?

2. Fig 2 was not exact manner, the remarketing line should be middle of the study and ES. Same for Fig 4 and 5.

Thanks for the study. Good Luck.

Reviewer #4: The paper is well written and the analysis is up to the standard. The PRISMA checklist is used and attached as per guideline of reporting a systematic review article. The discussion section is well argued and reference list is adequate. Congratulations team of authors

6. PLOS authors have the option to publish the peer review history of their article (what does this mean?). If published, this will include your full peer review and any attached files.

Reviewer #1: **Yes: **Shakil Ahmed

Reviewer #2: No

Reviewer #3: **Yes: **Dr. Ummul Khair Alam

Reviewer #4: **Yes: **Prof. Rose Mjawa Laisser

---

## [Author Response · Author response to Decision Letter 0]

13 Aug 2024

Date: August 11, 2024

Subject: Response to editor, and reviewers for the revised manuscript submission.

Title: Parental satisfaction towards care given at neonatal intensive care unit in Ethiopia: a systematic review and meta-analysis; [PONE-D-24-18838]

Dear Editor and Reviewers,

We are grateful for your careful reading of our work and your insightful comments. We truly value the time and energy you invested in offering thoughtful feedback and recommendations. Every one of your comments has been thoroughly reviewed, and the appropriate changes have been implemented in response. The individual reviewer comments have been addressed by explaining the adjustments made and offering a thorough response to each comment. We have also considered the journal requirements and ensured that our paper conforms to all relevant guidelines and formatting requirements. We believe that these changes have greatly improved our study's quality, precision, and clarity. We are certain that the revised manuscript now successfully addresses the issues brought up throughout the review process and complies with the journal's requirements. We would want to thank you again for all of your helpful advice and suggestions during this process. We welcome any further comments or advice you may have.

Sincerely,

The authors

Find my suggestions here below:

Indicate the PROSPERO Registration Number:

Clearly state the PROSPERO registration number for this systematic review and meta-analysis (SRMA) in the manuscript.

Response: We appreciate your insightful comment. We apologize for the missing PROSPERO registration number in the previous submission. The updated manuscript now includes the PROSPERO registration number (CRD42024570971) on page number 2 and 5, line number 33 and 110. We thank you for your valuable comment.

Use Appropriate Regional Nomenclature:

Ensure that the appropriate and correct naming of regions within the respective country is consistently used throughout the manuscript. This includes all text references, figures (such as Figure 4), tables, and any supplementary materials. for example Oromo Vs. Oromia, Harar Vs. Harari, Southern Vs. SNNPR etc.

Response: We value your meticulous attention to detail. We have carefully reviewed the manuscript to ensure consistency in the naming of regions. We have appropriately changed all text references, figures, tables, and supplemental materials in the amended document. We believe these changes improve the manuscript's quality.

Detailed Results Section in Abstract is needed:

Expand the results section of the abstract to provide more detailed and engaging information. Include key findings, statistical significance, and any important trends or patterns observed in the data. This will make the abstract more informative and appealing to the audience.

Response: We appreciate your insightful recommendation to make the results part of the abstract longer. We have meticulously rewritten the abstract to incorporate more detailed findings, including key results, statistical significance, and salient trends (page number 2, lines 39-41). We have tried to present a thorough synopsis of the study, but we have strictly followed the journal's 300-word limit. Thank you once again for your insightful suggestions.

Change Subtopic Title:

Change the title of the subtopic from "Result" to "Results" to reflect the content accurately.

Response: We appreciate your recommendation to change the subtopic title from "Result" to "Results." This modification has been made to correctly reflect the section's content in the revised manuscript. 

Reviewer #1: Thanks to the authors of the manuscript for trying to synthesize evidence on patient satisfaction in NICU. In my review, I have found some methodological issues and I have organized my feedback chronological to the chapters and subchapters of this manuscript. Please find the followings:

1) Abstracts

1.1: In the abstract's method section, if we have any word limitation then, I would prefer mentioning about the risk of bias of this SRMA, rather than mentioning which software we use to extract data or analyse the data.

Response: We are grateful for your recommendation to prioritize information in the abstract's methods section. Following the journal's rigorous 300-word limit, we have concentrated on an assessment of the risk of bias in the revised manuscript (lines 31-33). Thank you once again for your valuable suggestions.

2) Introduction

2.1: It is ambitious to say, this the first kind of study, I'd prefer rephrasing this sentence as this study might be the extension of the available evidence. Also, I just found one published SRMA on the similar topic from Ethiopia (https://www.ncbi.nlm.nih.gov/pmc/articles/PMC11000344/).

Response: We value the insightful reference you provided (https://www.ncbi.nlm.nih.gov/pmc/articles/PMC11000344) and your astute observation. The statement has been updated to reflect the previous studies. Our study expands on the body of knowledge by performing a more thorough systematic review and meta-analysis (SRMA), incorporating, a more comprehensive view of the data available compared to previous studies like the one mentioned, which included a small number of articles with a smaller sample size. 

3) Methods

3.1: Language Restriction: Including only English articles in the era of internet, AI, google translators, proves your literature search might miss some of the important articles published in other language.

Response: We really acknowledge your concern regarding the potential limitations of restricting the search to publications written in the English language. Although we can't rule out the potential of overlooking pertinent research written in other languages, a careful review of Ethiopia's body of literature on the subject turned up no research written in any language other than English (page number 7, lines 138-139). We used the available translation tools to perform preliminary searches in Ethiopian languages in order to further address your concern. There were no further related studies found by these searches. We believe that our thorough search strategy, including the exploration of non-English literature, sufficiently handles the possibility of any pertinent information being overlooked.

3.2: Exclusion criteria: Excluding articles that did not report on the prevalence of parental satisfaction is an inappropriate approach for this type of SRMA where the main focus is on patient satisfaction. Most of the studies use different scales to report patient satisfaction and they have different levels of reporting. Even for this particular research, all the included studies did not use the same tool to assess the patient satisfaction. I would suggest the authors to read Cochrane SRMA handbook and their published SRMA to have a clear idea how to report SRMAs with different scales and different reporting system.

Response: We appreciate your insightful suggestions on the exclusion criteria. After giving your idea great thought, we have updated our exclusion criteria in the revised manuscript (page number 7, lines 141-144). We have included all types of observational study designs in our study. To guarantee the validity of our study, we have followed the recommendations made in the Cochrane Handbook for Systematic Reviews of Interventions. Again, we are very grateful for your review.

3.3: Exclusion Criteria: How did the authors define, methodological problems, such as inadequate sample size, sampling techniques, incomplete data, inadequate statistical analysis, or any other relevant methodological limitations? What are their thresholds of exclusion? This needs to be mentioned in the Method section clearly.

Response: We thank you for your meticulous attention to methodological rigor. We have updated our exclusion criteria in the revised manuscript (page number 7, lines 141-144). In the updated manuscript, we have corrected typographical errors and believe these changes improve the soundness of the exclusion criteria.

3.4: The Holy 2012 tool can be used to assess the Risk of bias of the prevalence based articles, it doest not assess the internal and external validity of the articles. Please omit or rephrase the sentence.

Response: We appreciate and value your recommendation to make the Holy 2012 tool's scope clearer. The sentences have been changed in the updated manuscript (page number 8, lines 165-166). Thank you once again for your recommendation.

3.5: Why the authors assess Risk of Bias using two different scales (NOQAS and Holy tool) since authors only included the prevalence studies for this SRMA?

Response: We really appreciate your feedback and value your thorough assessment of our manuscript. Although the review primarily concentrates on the prevalence of parental satisfaction, it is crucial to remember that we also explored associated factors that may have an impact on the prevalence. We used the Holy tool especially to measure the risk of bias and the NOQAS tool to evaluate the quality of each study in order to completely analyze the quality of the included studies. The selection of NOQAS was based on its popularity as a tool for evaluating the quality of observational studies. Conversely, the Holy Tool was applied with greater specificity to articles that were prevalence-based. We were able to offer a thorough assessment of the included studies by making use of both approaches. We believe that using both approaches strengthens the quality of our findings.

3.6: Statistical analysis: Authors are reporting prevalence of satisfaction, it is unclear why authors have done the meta-regression here.

Response: We appreciate your thoughtful assessment of our statistical analysis and your comments. Our study explores associated factors that affect parental satisfaction in addition to the prevalence of parental satisfaction, which is our main finding. To investigate the effect of these characteristics on the prevalence of parental satisfaction across the included studies, meta-regression was utilized. In order to investigate possible sources of heterogeneity in the pooled prevalence, we also performed meta-regression. Once again, we are very grateful for your review.

3.7: GRADE assessment is an important tool to assess the certainty of evidence of your meta-analysis, which authors skipped. I'd suggest add this to the manuscript.

Response: We appreciate your insightful suggestion. We agree that GRADE assessment is important for determining the certainty of evidence in meta-analysis. We have updated the manuscript with a thorough GRADE assessment based on your suggestion. The GRADE assessment is now included in the methods section, statistical analysis subsection (page number 9, lines 187–190), where we provide a thorough review of the certainty of evidence for each outcome. We believe that this improvement greatly increases the transparency and rigor of our study. Once again, we are very grateful for your valuable suggestions.

4) Results

4.1: What did the authors do since they got very high heterogeneity? They did mention they explored their heterogeneity through subgroup analysis. However, in their forest plots, i did not find and I^2 value for the subgroups, which is very perplexing for the authors. They did report I^2 value in their "Results: Subgroup Analysis" section; however, I believe authors have interpreted the heterogeneity in their subgroup analysis incorrectly. The I^2 = 90.45% is the overall heterogeneity. Please follow the Cochrane SRMA handbook or other available literatures.

Response: We understand that the level of heterogeneity in our meta-analysis concerns you. Because of the diverse geographical and socio-economic situations across different regions of Ethiopia, it is expected that there will be variations in the outcomes. To make sense of this, we performed sensitivity analysis and meta-regression in addition to subgroup analysis to investigate possible sources of heterogeneity. However, the result for meta-regression was insignificant, and sensitivity analysis also indicated that no study had a significant impact on the overall pooled estimated prevalence as this is presented in our manuscript (page number 13, lines 262-269). In the amended manuscript, we have added I2 values for every subgroup analysis to the forest plots, as advised. In the event that our result had very high heterogeneity, we would write this as a limitation in our manuscript. Once again, we sincerely appreciate your contribution in bringing these concerns to our attention, and we value your efforts to improve the quality of our study.

4.2: For the forest plots, please try to add the I^2 values for each of the subgroups, chi-square statistics, and p-value.

Response: We thank you for your attention to detail and your suggestions on how to make our manuscript better. Your request to include chi-square statistics in the forest plots has been carefully reviewed. But as you correctly pointed out, our analysis includes 11 articles across five geographical regions of Ethiopia, with a maximum of four studies conducted in the Amhara region and a minimum of one study in the Southern Nations, Nationalities, and Peoples' Region. As chi-square statistics are usually used to evaluate heterogeneity among multiple studies within a subgroup, this particular feature of our dataset makes them challenging to compute. To address your concerns, we have added I² values for each subgroup and the overall p-value in the forest plots. However, the calculated I² values and p-values were (I^2 values =. % and p-value =.) as shown in the subgroup analysis (Figures 4 and 5). This is due to the limited number of studies (three or fewer) in each region or year. We would be pleased to offer more information or analysis if necessary, and we are still available for additional clarification on this topic. Again, thank you so much for your valuable feedback.

Reviewer #2: Review Reports

Title: Parental satisfaction towards care given at neonatal intensive care unit in Ethiopia: a systematic review and meta-analysis Manuscript Number: PONE-D-24-18838

Comments

• The issues is contemporary agenda.

Response: Thank you for recognizing the study's relevance to current research objectives. In NICUs across the globe, parent satisfaction with newborn care is a vital component of quality improvement. In particular, the Ethiopian context, where improving parental experiences is desperately needed, is the subject of this study.

• The background fails to entail all what it intends to entail. Avoid repeated use of sentences. E.g. Adverse consequences of dis-satisfaction.

Response: We appreciate your feedback. With great effort, the background section was created to offer a thorough synopsis of the pertinent literature. We believe the details provided are adequate to illustrate the background and purpose of the research. We have carefully examined the section and made the required changes to improve its coherence and clarity. We have also avoided the repetition of sentences in the revised manuscript. Again, Thank you for your insightful comment.

• The methods fail to describe whether the inclusion of unpublished articles and its where about.

Response: We appreciate your valuable feedback. This systematic review contained unpublished articles. We looked through the grey literature in great detail to find relevant studies. The search approach used to find unpublished articles is also clearly described in the methods section, together with information on the databases and sources that were looked for (page number 5, lines 106 and 113). We have carefully gone over this section to make sure all of the information is accurate and thorough. Thanks once again for your valuable comment.

• The extractors, ambiguity an dhow the ambiguity was solved as well as the inclusion of retracted publication is not mentioned.

Response: We really appreciate your comment. The data extraction process is described in detail in the methods section (page number 7, lines 146–151), together with information about the extractors' identities and ambiguity-resolution techniques (lines 125-128). We also clearly indicate that papers that were retracted were not included in the analysis.

---

## [Decision Letter · Decision Letter 1]

10 Sep 2024

PONE-D-24-18838R1Parental satisfaction towards care given at neonatal intensive care unit in Ethiopia: a systematic review and meta-analysisPLOS ONE

Dear Dr. Haile,

Thank you for submitting your manuscript to PLOS ONE. After careful consideration, we feel that it has merit but does not fully meet PLOS ONE’s publication criteria as it currently stands. Therefore, we invite you to submit a revised version of the manuscript that addresses the points raised during the review process.

We look forward to receiving your revised manuscript.

Kind regards,

Tamirat Getachew

Academic Editor

PLOS ONE

Reviewers' comments:

Reviewer's Responses to Questions

**Comments to the Author**

1. If the authors have adequately addressed your comments raised in a previous round of review and you feel that this manuscript is now acceptable for publication, you may indicate that here to bypass the “Comments to the Author” section, enter your conflict of interest statement in the “Confidential to Editor” section, and submit your "Accept" recommendation.

Reviewer #2: All comments have been addressed

2. Is the manuscript technically sound, and do the data support the conclusions?

Reviewer #2: Partly

3. Has the statistical analysis been performed appropriately and rigorously? 

Reviewer #2: Yes

4. Have the authors made all data underlying the findings in their manuscript fully available?

Reviewer #2: Yes

5. Is the manuscript presented in an intelligible fashion and written in standard English?

Reviewer #2: Yes

6. Review Comments to the Author

Reviewer #2: Review Report

Abstract: Needs further enrichment

Background:Relatively weak

Methods: Not comprehensive

Result and discussion: Is weakly logical and the explanation should be rich with appropriate references.

Regards,

7. PLOS authors have the option to publish the peer review history of their article (what does this mean?). If published, this will include your full peer review and any attached files.

Reviewer #2: No

---

## [Author Response · Author response to Decision Letter 1]

19 Oct 2024

Subject: Response to reviewer for the revised manuscript submission.

Title: Parental satisfaction towards care given at neonatal intensive care unit in Ethiopia: a systematic review and meta-analysis; [PONE-D-24-18838]

Dear Editor and Reviewers,

We appreciate your time and expertise in reviewing our manuscript. your valuable input has undoubtedly contributed to the enhancement of our manuscript. We appreciate your time and effort in reviewing the previous version and considering our responses to your comments. We are glad that the revisions we made in response to your previous comments have addressed your concerns and met your expectations. Once again, we sincerely appreciate your positive response and we are grateful for your thorough evaluation of our manuscript.

Thank you so much for your time and consideration.

Sincerely,

The authors

Reviewer #2: Review Report

Abstract: Needs further enrichment

Response: We are grateful for your valuable feedback for our abstract to make enrich. We understand that the abstract could be more detailed/enrich. However, we are unable to make enrich/expand the abstract section because of the journal's strict word limit (no more than 300 words) unless we intend to avoid some sentences. Thank you once again for your valuable suggestions.

Background: Relatively weak

Response: We appreciate your thoughtful feedback. We have made some modifications in the background section based on your suggestion by adding one paragraph that is detailed about the neonatal intensive care unit and preterm newborn (page number 3, lines 59-67). We believe the details provided are adequate to illustrate the background and purpose of the research. If you have any further comments, suggestions, or specific areas you would like us to add/focus on, please let us know. Thank you once again for your insightful comment.

Methods: Not comprehensive

Response: We appreciate your review and valuable feedback on our methods section. We worked hard to create a comprehensive methods section that provides a thorough overview of our manuscript. If you have any further comments, suggestions, or specific areas you would like us to focus on, please let us know. Again, we appreciate your insightful comment.

Result and discussion: Is weakly logical and the explanation should be rich with appropriate references.

Response: We believe that the logic connecting our results to our discussion is sound and well-supported. Additionally, the references cited within the results and discussion are also appropriate. Please do not hesitate to point out any particular examples, where you believe the logic is weak or where you believe more references are needed. We appreciate for you valuable input.

---

## [Editor Report · Decision Letter 2]

24 Oct 2024

Parental satisfaction towards care given at neonatal intensive care unit in Ethiopia: a systematic review and meta-analysis

PONE-D-24-18838R2

Dear Dr. Teklehaimanot,

We’re pleased to inform you that your manuscript has been judged scientifically suitable for publication and will be formally accepted for publication once it meets all outstanding technical requirements.

Kind regards,

Tamirat Getachew

Academic Editor

PLOS ONE
---

## [Editor Report · Acceptance letter]

28 Oct 2024

PONE-D-24-18838R2 

PLOS ONE

Dear Dr. Haile, 

I'm pleased to inform you that your manuscript has been deemed suitable for publication in PLOS ONE. Congratulations! Your manuscript is now being handed over to our production team.

Kind regards, 

on behalf of

Dr. Tamirat Getachew 

Academic Editor

PLOS ONE